# Internet of Things (IoT)-Based Teaching and Learning: Modern Trends and Open Challenges

**Ibrahim Ahmed Ghashim** [1] **and Muhammad Arshad** [2,*]

1. Deanship of E-Learning and Information Technology, Jazan University, Alburj Campus, Jazan 82812, Gizan, Saudi Arabia; ibrahimg@jazanu.edu.sa
2. School of Informatics and Cybersecurity, Technological University Dublin, Blanchardstown, D15 YV78 Dublin, Ireland
* Correspondence: muhammad.arshad@tudublin.ie

**Abstract:** The effect of technology has caused many institutions to intend to change their approach to teaching and learning, resulting in the current model of teaching and learning being an active collaborative and self-directed one. The connection between education and technology has received attention as part of educational policy and practice. Debatable topics like how to use technology in the classroom have therefore punctuated complaints about declining educational standards, unemployed learners, duplicate curricula, and archaic institutional structure. In the framework of information and communication technologies and societal growth, the Internet of Things (IoT) is asserting its vital position. Institutions may improve learning outcomes with the use of the Internet of Things by offering enhanced learning experiences, increasing operational effectiveness, and collecting real-time, actionable insight into student performance. The current state of the Internet of Educational Things (IoET) is examined from several educational perspectives in this article; a few of them are highlighted, and both of their established and potential educational benefits are discussed. Additionally, it provides in-depth discussions on current issues and problems for the IoET from a variety of approaches. The investigation performs a bibliometric analysis with VOSviewer to focus on peer-reviewed research articles published in well-known publications databases over the past eight (8) years. The findings of this study should also act as a strong incentive for universities and colleges around the world to use IoT-based teaching and learning technology for improved academic achievement.

**Keywords:** Internet of Things; emerging technology; education; modern trends; challenges

## 1. Introduction

The "God's Architect" Antoni Gaud introduced a fluid building style in 19th-century Spain by building them as three-dimensional scale models and shaping the details as he had them in his head. His buildings' expressive curves were not only functional load-bearing elements but also beautiful disguises. The "Internet of Things", also referred to as the "Internet of Objects", is a network of physical objects such as cars, appliances, and other household items that are connected and can exchange data. These objects are implanted with electronics, software, sensors, actuators, and connectivity. This type of system can be summarily described as a network of physical devices, vehicles, home appliances, and other items [1]. Each object has an embedded computing system that makes it uniquely identifiable, but they can all work together within the current Internet infrastructure. The vision of the Internet of Things, according to Wigmore [2], has developed as a result of the confluence of several technologies, including ubiquitous wireless connection, real-time analytics, machine learning, commodity sensors, and embedded systems. As a result, the traditional fields of embedded systems, wireless sensor networks, control systems, automation (including home and building automation), and others all assist in enabling the

Internet of Things. The Internet of Things (IoT) is transforming many facets of our daily lives. Due to their ubiquity and the encouragement of intelligent and autonomous solutions, IoT technologies vary from earlier advances [3,4]. IoT advancements are a significant strategic technology trend [5]. The conceptual base for the new learning model was thought to be ubiquitous sensors and the capacity to connect the physical and digital worlds. The capacity to integrate sensors into any object and use Machine-to-Machine (M2M) communication to connect billions of objects/devices to the existing Internet infrastructure is the idea underlying this major paradigm change. The physical world as a whole is quickly going online. The IoT is expanding swiftly and becoming a worldwide concern that inspires both enthusiasm and trepidation [6]. There are several signs that the Internet of Things will transform various industries, including higher education, particularly universities. Universities now have the chance to take the lead in IoT technical development and innovation models, develop future IoT leaders, and handle threats regarding TIPPSS, which stands for Trust, Identity, Privacy, Protection, Safety, and Security, concerning the IoT [7,8].

After discovering how crucial STEM is to boosting economies around the world, there has been a growing interest in the fields of science, technology, engineering, and mathematics (STEM). In Germany, where technology is ingrained in cultures, the fourth industrial revolution (4IR) was initially introduced. One of the key elements of 4IR is the IoT. Although there is a growing need for experienced individuals in the IoT, there are not many higher education institutions that provide STEM-related IoT courses. According to research, STEM students lack or have little experience in designing and implementing IoT applications [9]. As a result, such students enter the workforce later with little or no IoT experience. There is not enough space in higher education institutions' present STEM curricula for numerous courses relating to the Internet of Things. The majority of higher education institutions worldwide do not provide a curriculum to advance students' IoT knowledge and abilities. This exhorts teachers to include IoT technology in their lessons. IoT technology will, according to experts, be the most important and impactful technology in the following years. IoT is a broad umbrella term that may be used to group many different sorts of digital learning. For instance, E-Learning (Electronic-Learning) describes the type of learning that uses electronic resources including computers and networks (the Internet, Intranet, and Extranet) [10]. Any form of knowledge that is provided via handheld and portable devices is known as M-learning (Mobile-Learning) [11,12]. U-Learning (Ubiquitous-Learning) is a type of straightforward Mobile-Learning. The cast of U-learning, where students can access additional learning materials and make use of the resources of collaborative learning spaces, is a step toward the pervasiveness of information [13,14].

The literature states that several studies have purportedly been conducted on the application of the Internet of Things (IoT) in the educational field. The study in [15] assessed how the Internet of Things was used in teaching and learning. The study examined how few models have been used to adopt the IoT and how little the IoT has been used in learning. The few models and approaches that have been employed were presented for deeper understanding. It also demonstrates the gaps in the research to emphasize how the Internet of Things could aid with issues linked to learning. The IoT has many advantages, but innovations that address pedagogical and learning concerns have not yet been fully implemented. The IoT has many advantages for educational contexts that make it possible to track students' activity. As a result, the IoT enables educational institutions to swiftly handle student difficulties via study activities. Finally, this study demonstrates the advantages and disadvantages of using the IoT in teaching and learning. A study on the effects of the IoT on higher education, particularly universities, is presented in this publication [16]. The IoT has the potential to significantly change how colleges function and enhance student learning across a variety of courses and grade levels. If leadership, staff, and students are well-prepared to ensure widespread and successful implementation, it offers enormous potential for universities and other educational institutions. The IoT

needs to be developed, and institutions may take the lead. A special opportunity exists for academics, researchers, and students to pioneer the creation of IoT systems, devices, applications, and services.

The authors in [17] demonstrated that there are other ways that the IoT is used in higher education besides teaching and learning. The study, instead, provided a list of several IoT-related uses in higher education. In the first instance, cameras are used to control energy and monitor the environment. The second is a secure campus and classroom access control, where the IoT is used to provide secure academic settings for instructors and students. Third, the IoT is utilized to monitor patients and prevent diseases in the context of student health monitoring. Fourth, enhancing pedagogy by utilizing the IoT to improve teaching and learning. Additionally, the authors in [18] carried out a research study to see if IoT use in education benefits online teaching and learning. To achieve this objective, a system has been proposed that enables the administration of the educational institution to make knowledgeable decisions. The suggested system would gather data from IoT devices and analyze them to enhance the information that teachers and students would receive. It has been determined that IoT use in education benefits online instruction and learning. The authors in [19] studied the IoT smart campus paradigm and applications concerning education. The study's implementation on the campus of the university includes descriptions of these applications, which include an IoT-based flipped classroom, an IoT-based admission system, student feedback, an IoT-based orangery, and an IoT heating system. In a study on personalized learning, the authors in [20] used data gathered by the IoT, cloud computing, and learning analytics technologies to offer learners individualized education. According to reports, this strategy can deliver curricula that are personalized based on the skills of each student. By allowing students to interact with real-world objects that are virtually connected to a particular learning topic, the authors proposed a system to improve student learning. In many academic programs, including those in medicine, computer engineering, and mechanical engineering, students must interact with various things. The proposed technology supports instructors in their teaching process and improves student academic performance, according to tests conducted on the functional prototype.

The Intelligence of Learning Things (IoLT) is the name of the educational platform suggested by the authors in [21]. It is a blended learning approach based on the Internet of Things that can improve the traditional educational system with cutting-edge teaching methods and technologies. On this platform, a variety of tools and applications can be used by participants (such as professors and students) to collaborate and exchange ideas. Therefore, our suggested methodology can guarantee that you acquire wiser and more effective methods than the standard means. The goal of the paper [22] is to introduce the Internet of Things (IoT), a novel idea in IT&C (Information Technology and Communications). A study is undertaken to show the value of introducing the IoT in higher education, starting with a brief introduction to the fundamentals of this concept. Additionally, several doable ways have been identified for incorporating IoT elements in academic settings, particularly for improving teaching and learning. Although significant IT&C corporations have started and completed projects in this field, a model for "smart universities" has not yet been established. In our analysis, we show that IoT Platforms with real-time, limited-area service provision employing Cloud Computing services are the best technical answer for academia. The study in [23] provided a perspective on how both the IoT and big data could have positive effects on higher education. This study makes the case that big data and the Internet of Things (IoT) can both be used to enhance learning since they support intelligent connections in different ways. The result is a learner who is better-off thanks to an efficient learning environment. According to the study, these technologies also have difficulties with security, transparency, and vast amounts of acquired data. Therefore, this study focuses on the connection between the IoT and big data, explores their potential applications in education, and highlights their challenges to advance the field of education. The Internet of Things (IoT) is a network of numerous distinct "connected things" that is

quickly expanding. The use of the IoT in academia is a recent trend that has created new opportunities and possibilities for the infrastructure of educational institutions as well as the improvement of teaching and learning processes. Due to the demand for information in instructional technologies, the study in [1] suggests that institutions of higher learning should reduce latency time in their enterprise architecture. New approaches that respect each person's privacy, preferences, and expectations, while fostering the development of novel products and services, must be created. Innovative funding strategies for information technology infrastructure and services must be developed in higher education.

This study is looking into how cutting-edge technology is being used in teaching and learning. Since the teaching and learning environment will someday need to accommodate trillions of devices, software applications, and hardware components, the necessity for developing technologies is what drives this effort. This study found that the Internet of Things (IoT) allows for more individualized learning and teaching experiences, which has the potential to dramatically improve teaching and learning approaches. The architecture of the paper is organized as follows: An overview of the theoretical concept of IoT-based teaching and learning is explained in the next section, followed by a research methodology. Section 4 describes the modern trends of IoT-based teaching and learning; the open challenges are discussed in Section 5. The results and discussions are explained in Section 6, and lastly, the conclusion and limitations of IoT-based teaching and learning are presented in Sections 7 and 8.

## 2. Theoretical Concept

According to a common definition of the Internet of Things, it is a network of individually addressed objects (devices) that use Internet technology to locate and communicate with one another [24]. The Internet of Things, one of the new paradigms that has emerged as a result of the Internet's recent rapid growth, is one of information and communication technology's hottest and most intriguing topics [25]. Horizon Reports are regarded as reliable studies that look at the most recent breakthroughs in education and educational technology. In the context of substantial advancements in educational technology for higher education, one of the technologies in the 2017 Horizon Report that is projected to enable adaptation within 2–3 years is the Internet of Things paradigm [26]. The concept and principles of the Internet of Things need to be understood by students who are going to use such technology in their professional lives over the coming years, as well as employees who will always leap towards this technology. A learning environment that enables the practical application of theoretical knowledge within these fields needs to be created [27]. Additionally, the ability of modern systems to integrate has brought attention to the necessity of multidisciplinary work [28]. On the other side, educational materials developed using the Internet of Things' functioning principles aim to increase student involvement with content and improve learning settings [29].

Technology developers now have the means to create less expensive, more compact wireless systems that can be included in practically any kind of gadget [30]. The following three IoT components enable frictionless connections: Hardware consists of sensors, actuators, and embedded communication tools; Middleware is a term used to describe tools for on-demand processing and storage for data analytics; and a demonstration of fresh, user-friendly visualization and interpretation tools that are broadly accessible across several platforms and can be developed for a range of applications [24]. Incorporating low-power communications into an Internet of Things node can be carried out in several ways, from specifically built protocols like ZigBee to low-power variations of Bluetooth, Wi-Fi, and NFC. The Internet of Things (IoT) enhances existing formats like Radio Frequency Identification (RFID) technology, which is a technology utilized in a variety of commercial, industrial, and personal technological systems that allow for the creation of microchips for wireless data transmission. Wi-Fi is the most widely used integrated wireless technology and has the highest power-per-bit transmission efficiency. Some of this technology enables

the addition of wireless sensor networks (WSN) to a variety of gadgets, including books and wearable fitness trackers like FitBit [16].

The Internet of Things (IoT) has altered traditional teaching methods, but it has also led to changes in university foundations. Internet of Educational Things (IoET) is a term used to describe using IoT tools to improve academic, teaching, and campus infrastructure in universities, as illustrated in Figure 1. When instructors use smart devices to govern and manage the classroom, they should be able to select when to talk louder as soon as their pupils start to disengage or their attention span starts to wane. Smart classrooms with cutting-edge Internet of Things tools can be used in intelligent settings as a class. The definition of "smart classrooms" was expanded to encompass the use of Internet of Things technologies by professors, such as RFID, cameras, a smart classroom roll call system, and sensors, to efficiently assess students and conduct the courses [31]. The following research questions served as the guide for the multi-phased search, evaluation, and analyses of academic articles on the IoT for this study:

1.    What does the IoT (Internet of Things) technology for education mean?
2.    What are the advantages of education, according to publications from recognized research?
3.    What are the most recent developments in the Internet of Things (IoT) for education and what challenges remain?

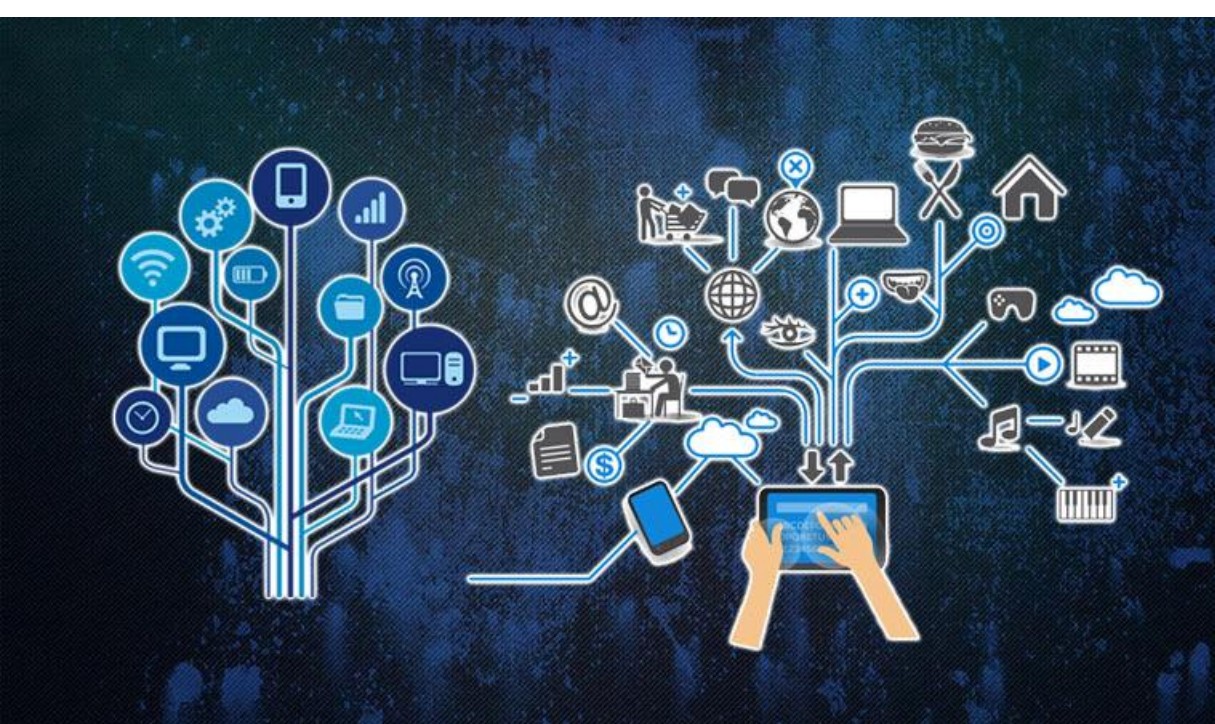

**Figure 1.** Internet of Educational Things (IoET) [32].

As a result of the pandemic phase, more educators now have the opportunity to test out IoT projects linked to protection, diagnosis, tracking, and lock-down. Such studies are anticipated to be essential in assisting educators in recognizing the benefits of IoT technology and actively implementing them in school environments.

## 3. Research Methodology

There are a lot of articles and conference proceedings being published in journals about the Internet of Things (IoT), which is a relatively new subject in education. To provide more innovative and engaging teaching tactics and educational opportunities, the IoT is being implemented in classrooms in the education sector. Teachers and administrators may expedite several time-consuming administrative activities like attendance tracking and

grading with IoT technologies. It may allow educators to devote more time to teaching and learning while concentrating on their primary responsibilities.

### 3.1. Selection of Database

To find a thorough overview of the relevant literature, we looked for articles in the following reputable databases: Scopus, WoS (Web of Science), Google Scholar, IEEE Xplore, and online sources. A total of 234 academic papers and publications were used in the research. The content of the chosen articles was then synthesized after removing duplicate articles that had been collected across several databases. Thus, as shown in Figure 2, we focused our search to include books that discussed "Internet of Things (IoT) based teaching and learning". Scopus is one of the two main commercial bibliographic databases with scholarly content from almost every field. Scopus provides author profiles, academic journal rankings, and an h-index calculator, in addition to research article searches. The second largest bibliographic database is the Web of Science, commonly referred to as the Web of Knowledge. On their campus network, academic institutions typically offer free access to the Web of Science or Scopus. The premier academic database for engineering and computer science is IEEE Xplore. In addition to journal articles, one can search for conference papers, standards, and books [33,34].

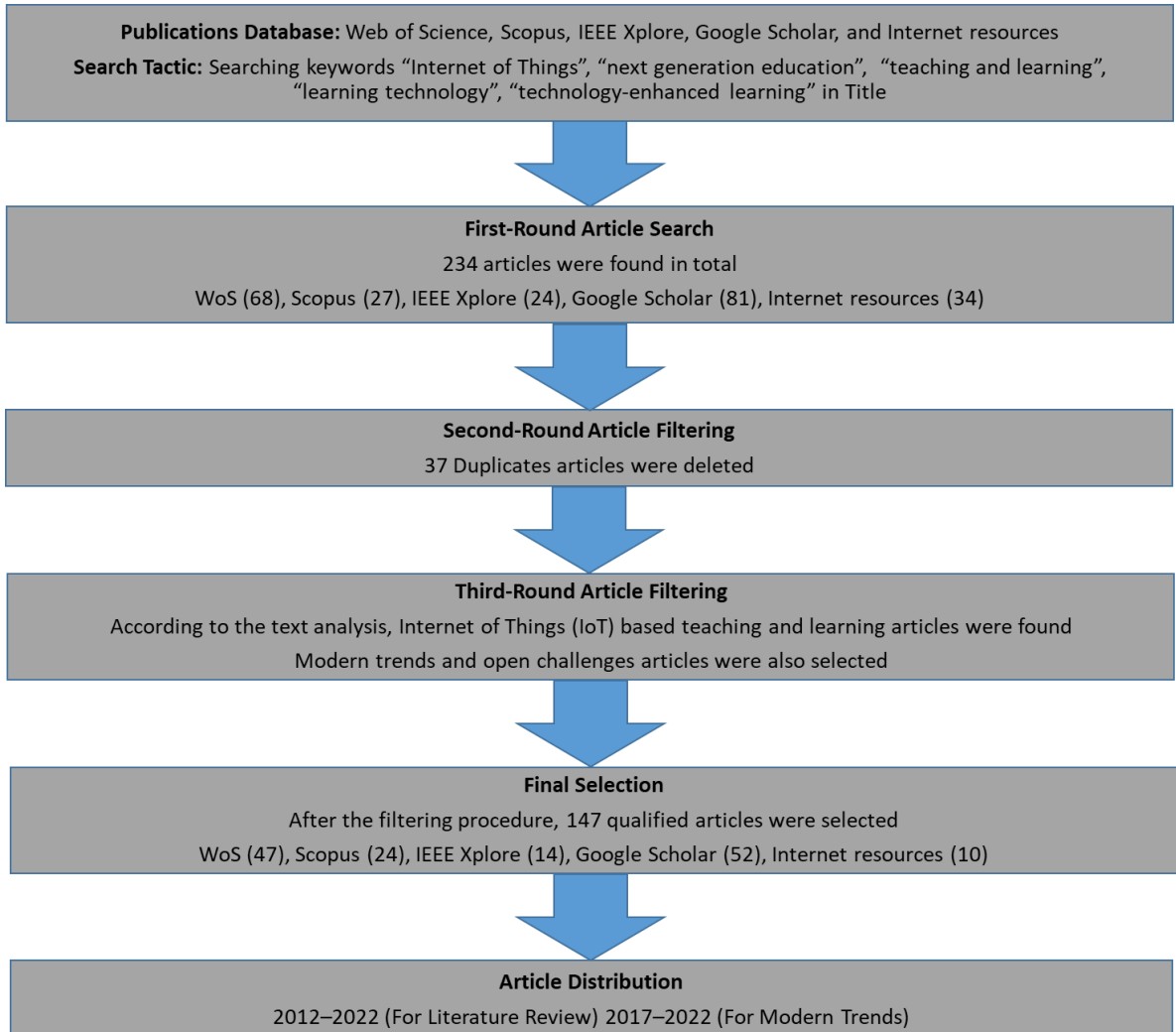

**Figure 2.** Classification of Research.

To conduct a comprehensive examination of the literature for the current investigation, the researchers used a variety of keywords and addressed them in the following sections.

### 3.2. Selection of Keywords

Peer-reviewed papers on cutting-edge educational technology are critically assessed in this multi-phased study. Multiple-phase searches and selections were conducted to find publications that could be subjected to in-depth analysis. Even with carefully stated criteria, it is impossible to do a thorough search given the abundance of online publications and open-access content. This study was designed expressly to concentrate on scholarly works from numerous well-known online sites. As newer journal papers might not be included in databases, additional search efforts were conducted to find more current publications in future or innovative education-specialized journals. Multiple keyword combinations were used to constantly search the source database, search strategies, and search phrases including "Internet of Things (IoT)", "next generation education", "teaching and learning", "learning technology", and "technology-enhanced learning". Additionally, similar searches were conducted for papers published up to 2023 on websites for research and development.

Bibliometrics can be employed as a tool to pinpoint areas of strength in research and provide guidance on potential directions for future study. To trace the intellectual history of a particular area of study, bibliometric analysis is an invaluable tool. It permits carrying out a more systematic literature study, gathering data, and spotting trends. Bibliometric networks can be built and visualized using the software program VOSviewer 1.6.18. These networks may be built via citation, bibliographic coupling, co-citation, or co-authorship relationships, and they may contain journals, researchers, or individual articles, for example. The keywords used by authors and bibliographers to create publications have been investigated using VOSviewer. Researchers could use VOSviewer to create maps for the research items. Keywords for a country or author are examples of themes that pertain to the study. Any two things might have an effective link. Each link has a strength, which is denoted by a positive integer; the higher this value, the stronger the relationship [35].

### 3.3. Filtering of Articles

To achieve the specified research goals, a set of inclusion and exclusion criteria were developed. The largest publication databases were only used to choose peer-reviewed, English-language journal publications that presented empirical, fact-based research for further analysis. Peer-reviewed journal articles, which effectively subject an author's work to the examination of other subject matter experts, have become the cornerstone of the scholarly publishing system. Since English is the lingua franca of science, it is the language that can most successfully traverse international borders and expand the influence of research. You must be proficient in English if you want to study abroad or promptly read recently published scientific papers. As a result, one can use the articles whenever they are required for their research projects and not just when they are available in their mother tongue [36]. As a result, it motivates authors to work hard to generate superior research that will improve the discipline. Additionally, the following standards are strictly followed during the screening and selection procedure [37]:

1. Research must be conducted on teaching and learning environments based on the Internet of Things. As a result, research that has been published on cutting-edge technology in fields other than education, such as engineering, the consumer market, healthcare systems, and others, has been ignored.
2. Empirical study using data is a requirement for research. Articles that relied heavily on personal tales or viewpoints were not eligible.
3. Through the presentation of significant qualitative data, research must evaluate how the Internet of Things (IoT) impacts education. Papers that provided no evidence of learning were rejected.
4. Theoretical, conceptual, and literary reviews were added in additional publications for thorough analysis. These publications were carefully examined to increase our theoretical foundation knowledge and to build a broader understanding of new technologies in teaching and learning.

*3.4. Techniques for Data Collection and Analysis*

The primary study is built on the primary literature, which consists mostly of scientific papers indexed in the top bibliographic indexes. Since bibliometric approaches allow for a full examination of publications on "Internet of Things (IoT) based Teaching and Learning" at various levels, they were employed as the foundation for the research methodology in this study. The suggested methodology is predicated on a quantitative examination of all relevant publications chosen to utilize keyword searches in the paper titles. The researchers examined each article that met the criteria, scrutinized it, and made the following conclusions: Bibliographic data, study countries, study locations, and technological trends in education. The researchers incorporated bibliometric data on the publication (such as the publication year, journal name, etc.), nations where studies were conducted, the educational setting of the study, and other data (such as K–12 or higher education). The researchers debated, examined, and categorized any repeating topics that emerged throughout the process, to conclude. Due to the use of a collaborative methodology in many studies, the review was very reliable and defensible [38,39].

## 4. Modern Trends of IoT-Based Teaching and Learning

The pace of schooling is accelerating due to the introduction of new technology and the emergence of generations who are tech-aware. IoT-enabled education solutions, including interactive displays, digital boards, language laboratories, tablets, and school security software, are essential for meeting the demands of these children as well. By enabling educational institutions to become Wi-Fi-enabled smart learning environments, the IoT is revolutionizing the education sector. The whole integration, intercommunication, and synchronization processes in modern smart systems can now be enabled via Wi-Fi and sensor technologies. Increasing Internet accessibility at the community level has always been difficult, but thanks to the IoT in education, we can alter classrooms even further and make technology use easier even in rural places.

The purpose of the study in [40] is to clarify how the IoT is being used to develop the Smart Education System, as discussed in Table 1. The goal of this study is to learn more about these interactions in contexts related to the smart physical campus. The IoT can be used to create new learning opportunities and scenarios for students, teachers, and other team members, enhancing their educational experience as illustrated in Figure 3. The study's drawback is that it primarily addresses the application components of the subject matter, leaving out the physical components. The Internet of Things has advanced capabilities that make it possible to apply e-learning after an architecture has been developed, and the article [41] analyzes the current and projected state of the IoT world as it relates to the subject of education. Real-time measurements of students' skin resistance, brain waves, and pulse rates are carried out by a group of IoT sensors, which also include cameras, microphones, and wearable gadgets. Utilizing the suggested design, institutions can modify their remote learning strategies to increase efficiency and make the most of their resources without altering their total scholarly activities. The study's findings suggest that e-learning has statistically significant positive effects on students' flexibility, learning experience, educational productivity, and overall quality of education. The article [42] gives a case study of how Prairie View A&M University (PVAMU) integrated cutting-edge IoT technologies into its Computer Science (CS) program. To gain students' interest in smart IoT, this article offers a collection of IoT learning modules that are simple to incorporate into already-existing CS curriculum courses. At PVAMU's CS department, the newly developed modules have been used to introduce a brand-new project-based course with a focus on intelligent IoT technologies. Also presented are the results of the curriculum change's external review. These show encouraging effects on IoT interest and knowledge throughout numerous courses and semesters.

**Table 1.** Summary of Selected Recent Publications between 2017 and 2022.

| Author(s) of Research | Year | Research Approach | Inventions/Findings/Results |
|---|---|---|---|
| Maksimovic [43] | 2017 | Green IoT Platform | This paper analyzes the possibilities of G-IoT utilization in engineering education. |
| Majeed and Ali. [44] | 2018 | Sensors and Wearable Devices | This research focuses on creating smart rooms, and smart parking as well as delivering smart education to students. |
| Kamal et al. [45] | 2018 | STEM Education | The goal of this project is to create a learning tool that can offer appropriate instruction to comprehend the idea of IoT and blockchain technology. |
| Henseler [46] | 2019 | Digital Forensic Tools | The Internet of Things Forensic Lab (IoT Forensic Lab) was established by the research team at The Hague Security Delta (HSD) Campus to enhance education. |
| Ramlowat and Pattanayak [47] | 2019 | Applications of IoT Systems | The review of IoT implementation in the sphere of education and its difficulties are the main topics of this study. |
| Vinaya Chandra and Krishna [40] | 2020 | The IoT in the Development of Intelligent Education System | The objective is to evaluate the potential IoT benefits for education and how the sector may be leveraged while addressing its problems and lowering associated risks. |
| Shrestha and Furqan [48] | 2020 | IoT Devices and Sensors | The research focuses on enhancing the online learning and teaching experience through the implementation of the IoT using available devices, sensors, and other technologies such as machine learning and artificial intelligence. |
| Siripongdee et al. [49] | 2020 | Blended Learning | This model's framework links six modules, a collection of databases, two types of contexts (classroom and personal), and two user interface roles (teacher and student). |
| Siripongdee et al. [50] | 2020 | Smart learning environment | In this study, BL was classified into three types of learning environments—digital, embedded, and side-by-side cases—each of which was further subdivided into four characteristics: F2F, self-paced, Tele-D, and ubiquitous. |
| Zhuang et al. [51] | 2021 | IoT platform tracking student activities and behaviors | This approach chooses the most appropriate learning resources and lessens the chance of exam fraud. |
| Liu et al. [52] | 2021 | An IoT-based wisdom education platform | The proposed platform's testing findings demonstrate its viability and demonstrate that it has high throughput, low application latency, and good practicability while being able to efficiently monitor classroom use. |
| Ahmed et al. [42] | 2022 | Integrating innovative IoT technologies | This study project introduced an experiential learning strategy, which promoted innovation. |
| Setiawan et al. [53] | 2022 | IoT technologies in Augmented Reality and Virtual Reality | This paper discusses the application of intelligent learning environments to smart city government procedures. |

**Table 1.** *Cont.*

| Author(s) of Research | Year | Research Approach | Inventions/Findings/Results |
|---|---|---|---|
| Zhang [54] | 2023 | IoT and Neural Network algorithms | This algorithm helps teachers by enabling them to more effectively carry out their teaching duties and evaluate and assess students' English translation skills. |
| Mahapatra et al. [55] | 2023 | The IoT-based gamified educational method | The suggested educational strategy makes use of field techniques and the gamification of course modules in an IoT environment for enhancing learning stages. |
| Mudassir et al. [56] | 2023 | IoT-based Classroom management | The purpose of this essay is to clarify how classroom management works and how it might raise academic standards. |

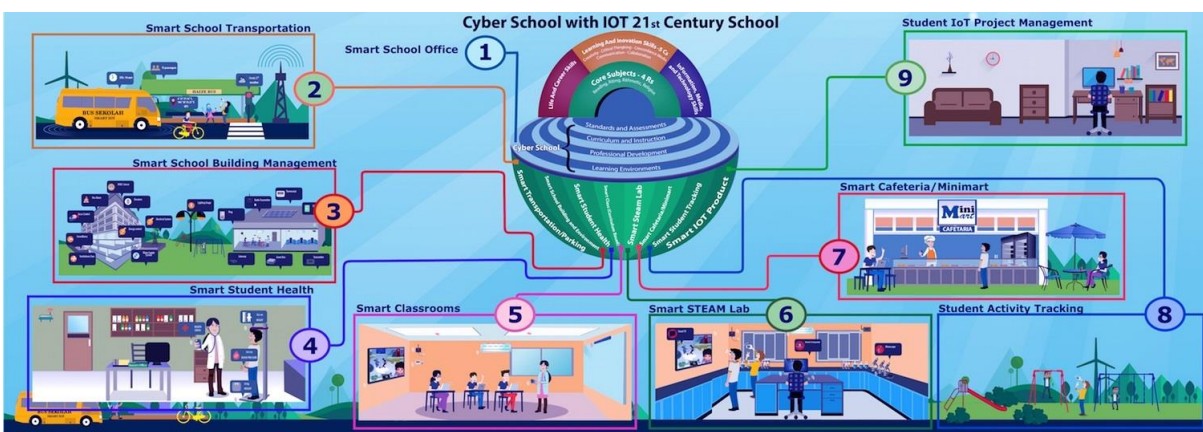

**Figure 3.** Roles and Applications of Smart Education [40].

The Internet of Things technology has been used to develop the university intelligence education platform [52]. The platform carries out the tasks of online instruction, attendance control, and examination result inquiry. The Internet of Things-based intelligent classroom architecture is created, and the specified classroom's environment equipment controller receives the intelligent classroom instruction. IOT technology can be utilized, develop the function module and cloud service layer of intelligent education information, and complete the design of the IoT-based intelligent education platform. The platform test results demonstrate that the throughput is higher, the application delay is reduced, and the platform's classroom assignment results are better. The outcomes of the experiments demonstrate the platform's strong application effect and dependability. The application of smart learning environments (SLEs) [53] in smart city governance processes is illustrated in Figure 4. To continue developing information based on the needs of e-learners, the researchers proposed the Internet of Things Virtual e-Learning System (IoT-Ve-LS) paradigm as a new Smart e-learning Tool (SeT). The suggested IoT-Ve-LS paradigm is a vast network, and connection to the Online Tutors (OTs) is crucial for e-learners. To accept, store, and distribute data in a cloud database, IoT-Ve-LS aims to protect a framework. The report split instructs the business on how to improve the intrinsically motivated online learner. Theoretical and fast local studies demonstrate the necessity and impact of the IoT in cutting-edge SLE, and the e-learning process is predictable and should not be ignored. Without regard to the current challenges in educational development, the IoT-Ve-LS's genuine and advantageous real-time scenarios are taken into account, providing a new evolution in teaching ecological systems for smart cities (SCs) in the future. Technology [48] has an impact on how people live, work, learn, and play. It has been essential in terms of education. The IoT refers to several devices and technologies that work together, and it transforms the way that

education is carried out. Using online IoT tools like Kahoot and communicating grades with students via Google Docs and Telegram can help teachers grade students more rapidly. Teachers give their kids access to informational Internet resources. Teachers' competence and knowledge can also be evaluated, and management will look into the matter, monitor events, and take the appropriate action in light of the feedback provided. In institutions without IoT devices, there are additional concerns with staff and student management, security, and excessive energy, heating, and water usage.

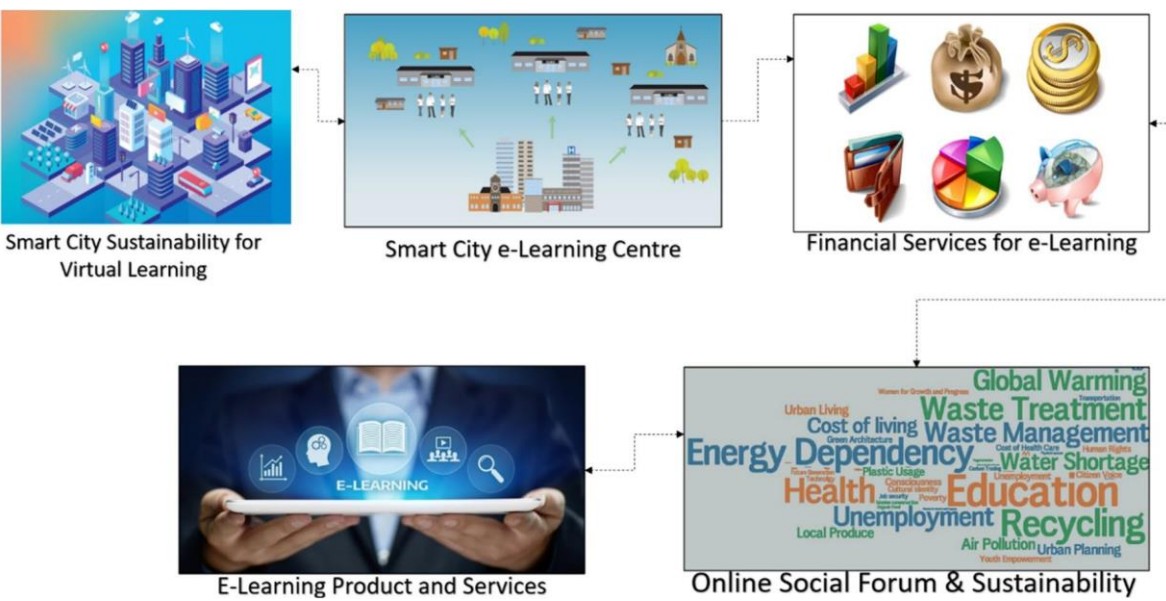

**Figure 4.** Setup of Smart Learning Environments Functions [53].

The Internet of Things Forensic Lab (IoT Forensic Lab), outfitted with specialized hardware, software, and expertise, was developed by the research team at The Hague Security Delta (HSD) Campus to improve education, as discussed in Table 1. A chip-off is an excellent illustration of a specialized instrument that may be used to retrieve data from memory devices, even if they are password-protected or have been damaged by fire or water. Teachers supervise bachelor students using cutting-edge digital forensic technologies in the IoT Forensic Lab while they are doing their internship [46]. A learning kit was created as a result of this research [45] to incorporate blockchain and Internet of Things (IoT) principles into STEM teaching, as discussed in Table 1. Through the use of a project-based learning strategy, the learning kit gives students practical experience. Parts for the "brain", "muscle", and "cloud" are included in the kit to cover all facets of blockchain and IoT technology. It is essential to present the concepts of the IoT and blockchain to pupils to teach them about the fourth industrial revolution. During the COVID-19 epidemic, a blended learning (BL) paradigm with an IoT base might be the best New Normal answer for all stakeholders in the education sector. Traditional face-to-face (F2F) is being pushed to change to stop COVID-19 because of social distance. Blended learning was categorized in the study in [50] into three different learning environments: digital, embedded, and side-by-side situations, as discussed in Table 1. These settings were then broken down into four categories: face-to-face, self-paced, tele-learning, and omnipresent. The content analysis method was used to examine and synthesize a model utilizing the pertinent literature, textbooks, research, articles, and websites as illustrated in Figure 5. This model's structure links six modules, several databases, two kinds of settings, and two different kinds of user interfaces (teacher and student) for usage in the classroom and on personal devices. The issues surrounding green IoT methods in engineering education to construct smart classrooms [43]. Engineering education has a strong emphasis on IoT resource sustainability, which motivates all participants in the

institution to operate responsibly, as discussed in Table 1. Several tasks must be finished, according to the author, to accomplish this. Making the most efficient use of resources, removing, reusing, and recycling IoT materials, and educating stakeholders at educational institutions about green IoT are some of these responsibilities. By achieving these goals, IoT resources created in the future would be more sustainably produced.

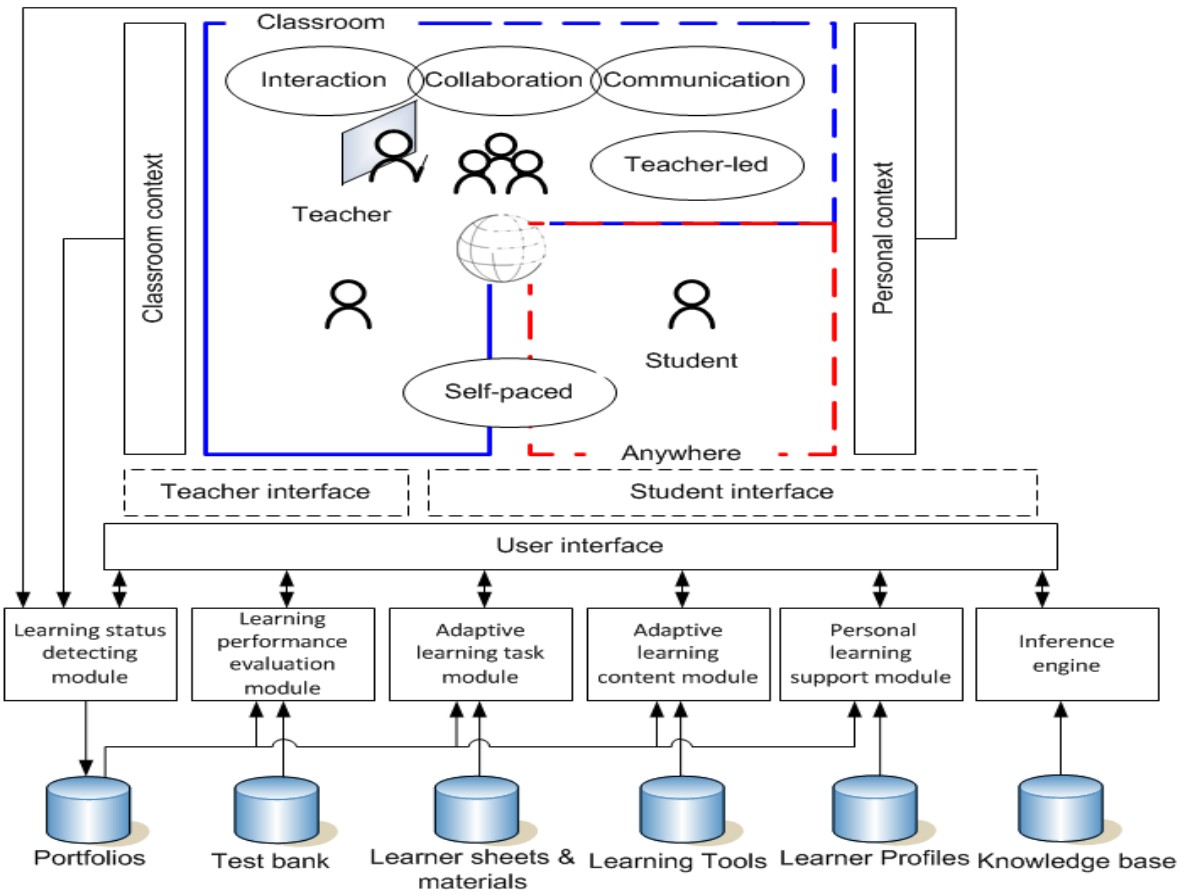

**Figure 5.** The Framework of a BL Model with IoT-based Technology [50].

In today's interconnected world, especially in nations like China where there is a greater premium on English language ability, English language translation education is essential. The assessment of students' translation skills and successful teaching, however, continue to be major obstacles for educators. To address the difficulties associated with teaching and translating the English language, this paper suggests a novel method that combines the Internet of Things (IoT), Particle Swarm Optimization (PSO), and Neural Network algorithms. This method will help teachers to carry out their teaching duties more effectively by assisting them in evaluating and assessing students' proficiency in translating the English language [54]. Due to the social and economic challenges they have faced in their lives, students have varying levels of learning capacity, as discussed in Table 1. Therefore, it is crucial to disclose the standards by which their learning capacity might be enhanced. This study explores these characteristics and seeks to enhance student learning skills through the use of a cutting-edge gamified educational approach, as illustrated in Figure 6. In this cutting-edge gamified instructional approach, the Internet of Things (IoT) is also used to gather data to construct a communication network and educational network. To improve learning phases, the suggested educational methodology used field methodologies and the gamification of course modules in an IoT environment, as discussed in Table 1. Engineering students are used to test the proposed educational approach throughout two courses [55]. The purpose of the paper in [56] is to clarify how classroom management works and how it might raise academic standards. This study employs

quantitative approaches and data analysis strategies while leveraging studies on literacy from journal papers on Google Scholar, as discussed in Table 1. To achieve school goals and create an environment that supports learning activities through student IoT (Internet of Things) and produces outputs that can benefit the community, the objectives of managing and utilizing existing resources, such as teaching staff, educators, and infrastructure, had to be completed. The quality of students and the quality of the classroom can both be raised by the effective management of student IoT and resources in schools. Teachers and students have learned the effectiveness of managing school resources by designing classroom management using IoT-based learning. Table 1 summarizes the selected findings proposed by different authors between 2017 and 2022.

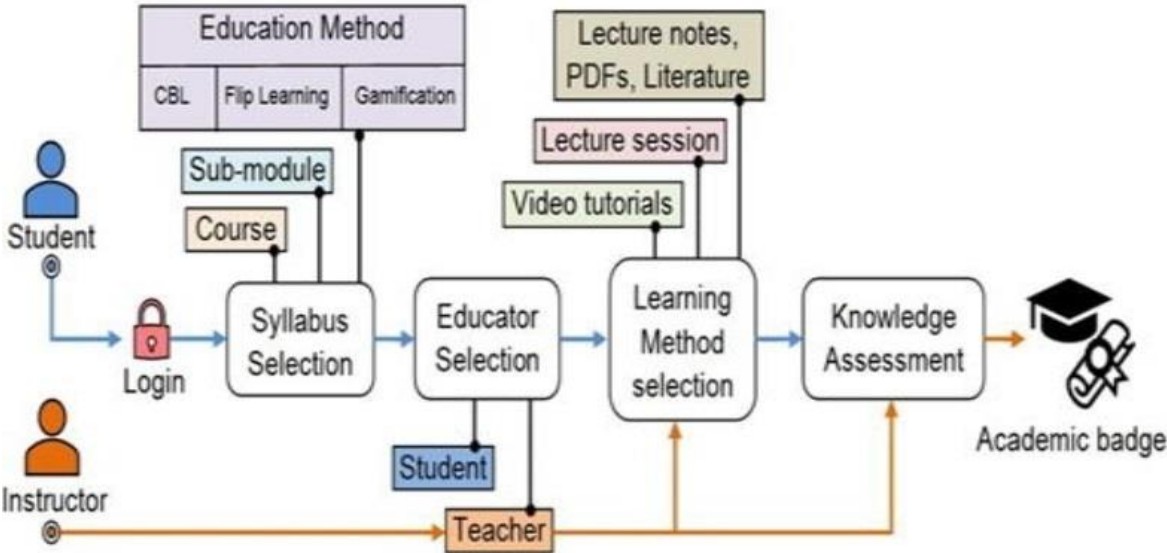

**Figure 6.** Game-based Education System through IoT Components [55].

## 5. Open Challenges of IoT-Based Teaching and Learning

The IoT can be a useful educational tool for students who use technology to solve unsolvable problems to increase their training skills. Students' academic behavior can also be tracked and recorded via wearable technology, which improves their interactive learning environment. Using the IoT in education, however, presents several challenges [57]. Departments must modify their curricula to include IoT courses to equip their recent graduates with the abilities needed to manage and work on a variety of IoT projects. The benefits of the IoT should be incorporated into developed curricula, staff should attend orientation sessions on the topic, teachers should have access to professional development opportunities, and students should be made aware of the different IoT uses. The Internet of Things is still in its beginning stages, and further challenges like wireless coverage, expensive sensors, and battery life have not yet been resolved [58]. IoT engineers and developers must take these issues into account to make it simpler for users to use IoT applications. Even though M-Learning applications like augmented reality and learning analytics have been the subject of various IoT-based studies, their adoption still needs development and more study. Concerns about security and privacy, according to many academics, are among the largest barriers to IoT adoption in education. Therefore, future efforts to implement the IoT in education must consider these factors to decrease the challenges observed in past studies and retain its effectiveness.

When our systems are connected to the Internet, there is a larger risk of cyberattacks. If any of the school's systems are targeted by students or outsiders, the entire system could be directly threatened. These efforts have the potential to render the institutions useless. A case of Internet failure might occur, rendering the entire educational system inoperable. Data theft is another major risk. IoT devices would capture a lot of data, therefore if these

data were stolen, it would be extremely dangerous to the security of the system. End-to-end security, authentication, and data confidentiality are among the other security and privacy issues that need to be addressed [59]. Only when they are trustworthy and secure can IoT applications be used in educational settings. IoT tools and applications use RIFDs and other devices as well as NFC, 4G, and 5G technologies to collect enormous amounts of data. Data security is, thus, a major concern. For IoT applications in the education sector, scalability is necessary. The IoT generates a lot of data, so data analysis is essential to gather a thorough understanding of the data. Because of this, scaling data in IoT applications for educational purposes is difficult. Sustainable IoT-based solutions should be made available to everyone [60]. Because of this, it must be offered to people right away. The systems of schools and other educational institutions must be able to afford these services and apps to run sustainably. Here, dehumanization refers to downplaying the significance of people. When we have autonomous systems that require little to no human interaction, dehumanization occurs. Dehumanization is one major problem that would develop if systems operated autonomously with less human interaction, giving service providers more authority and control [61]. Modern technology plays a significant role in our daily lives, but ethical concerns about dehumanization should be taken into consideration. For services and apps, an operational protocol is based on the IoT.

Listed below are a few more issues with the IoT in teaching and learning, along with some possible solutions. In particular, for cash-strapped educational institutions, the initial cost of obtaining and deploying IoT devices and systems might be significant. To improve the educational technology infrastructure, it is suggested to seek funding through grants, collaborations with tech businesses, or government programs. There is infrastructure and technology in place at many educational institutions. These outdated systems can be difficult to integrate IoT solutions with. The school's network infrastructure may become overburdened by IoT devices, which could cause connectivity problems. Upgrading the network's hardware will enable it to handle the increasing data load. Employ QoS (Quality of Service) and load-balancing techniques to guarantee uninterrupted connectivity. Internet of Things devices can break down or need routine maintenance, disrupting instruction and learning. Create a thorough maintenance and assistance plan. In addition, predictive maintenance using IoT to detect and address issues before they lead to disruptions. The use of the IoT to track student behavior and performance might lead to ethical questions about data use and surveillance. Data use and privacy rules should be clearly defined. Ensure that data collecting and monitoring are open to the public and solely utilized for educational purposes. When implementing IoT, educational institutions must adhere to several rules and norms, including data privacy legislation. To protect student data, keep up with pertinent rules and put the required compliance mechanisms in place, including COPPA (Children's Online Privacy Protection Act) or GDPR (General Data Protection Regulation). A lack of equal access to IoT devices and technology among students could result in unequal learning possibilities. Make sure that every student, regardless of socioeconomic background, has access to IoT devices in schools. Collaborate with community organizations to close the digital gap. Scaling IoT systems to accommodate more professors and students, as educational institutions expand, might be difficult. By selecting adaptable and expandable IoT solutions that can expand with the institution, one can start planning for scalability from the very beginning. It takes careful consideration, cooperation from different stakeholders, and a dedication to data security and privacy to address these issues. The IoT can dramatically improve teaching and learning in educational institutions when used appropriately.

## 6. Results and Discussion

This study made an effort to give a broad overview of current trends and unresolved issues in using IoT-based teaching and learning and its applications in diverse fields. Attempting to maintain a barrier between the academic setting and the most cutting-edge, contemporary technologies implies a significant loss for education in terms of communica-

tion and collaborative activities, as well as learning. More and more students refuse to use paper documents in favor of smartphones, tablets, and computers, which provide them access to the necessary information at their fingertips and the flexibility to learn at their own pace. Due to the increased effectiveness and student-centeredness of instructional techniques, this trend also benefits teachers. Teachers can concentrate on providing individualized care and attention to each student while collaborating with them via IoT-connected devices to modify their lessons and hands-on assignments. The IoT offers educational institutions, teachers, and students a variety of chances in general.

The Internet of Things (IoT) enables students to access educational materials from any instrument or gadget that is linked to the Internet. Additionally, by analyzing the data gathered from students via sensors and wearable technology, the IoT enables educational institutions to track and analyze students' academic progress [40]. Additionally, thanks to the Internet of Things, it is possible to identify both teachers and pupils, and attendance can be tracked automatically using RFID tags or face recognition software. Additionally, the IoT makes it possible for students to identify free study spaces and access additional classes whenever they need them by using occupancy detection and tracking features. In the article in [52], the implementation of smart learning environments (SLEs) in smart city governance procedures is examined. The Internet of Things Virtual e-learning System (IoT-Ve-LS) paradigm was developed by the researchers as a new Smart e-learning Tool (SeT) to continue developing material depending on the demands of e-learners.

The major goal of this study is to identify current trends in the use of IoT applications, platforms, smart learning environments, gamified education systems, sensors, and wearable devices in education and to present potential problems for future research. The IoT is a broad phrase that refers to a wide range of intelligent devices, apps, and services, as well as a massive amount of media resources. The IoT undoubtedly has numerous benefits for education and will continue to support learning processes, but we also need to draw attention to its drawbacks in terms of how it affects young people's education and guiding principles. Additionally, it is advised that teachers engage in online discussions and activities with their students, but it is advisable to warn them about the risks involved, such as inaccurate and imprecise information and plagiarism issues. Table 2 summarizes the Strengths, Weaknesses, Opportunities, and Threats (SWAT) analysis that is derived from this study.

**Table 2.** SWAT Analysis.

| Strength | Weaknesses | Opportunities | Threats |
| --- | --- | --- | --- |
| Enhanced Interactivity | Privacy Concerns | Improved Learning Outcomes | Data Security Breaches |
| Personalized Learning | Cost | Innovation | Technological Obsolescence |
| Efficiency | Security Risks | Accessibility | Digital Divide |
| Real-world Application | Technical Challenge | Global Collaboration | Resistance to Change |

Student performance can be greatly enhanced in several ways by the employment of cutting-edge hardware, software, and sensors in teaching and learning. These technologies improve learning by giving students more interactive, individualized, and data-driven educational possibilities. Modern hardware and software can adjust to the learning style and pace of each student. Interactive learning is possible with the use of tablets, interactive whiteboards, and virtual reality. Software and sensors can give students instant feedback on their performance. Gamification components are frequently used in educational software, which makes learning feel game-like. Modern hardware and software can assist in closing the gap between in-person and digital learning, especially in the context of remote or online education, ensuring that students can access top-notch instruction, even when they are not physically present in a classroom. With the use of technology, students may now easily access a wide range of online materials, such as simulations, instructional videos, and

textbooks. Analysis of student performance and the identification of potential problem areas can be conducted using data gathered from various sensors and software. With the use of technology, students can more easily collaborate on projects and homework. To ensure that students with disabilities have equal access to educational opportunities, devices and software can be modified to suit them. Students can undertake research and conduct in-depth topic exploration using the Internet and specialized applications. Students can manage their time and work properly with the support of digital tools and apps, which is essential for academic achievement. It is essential to remember that the efficiency of these technologies in raising student achievement may vary depending on elements like the standard of the educational content, the preparation of teachers, and the accessibility of these resources to the students. To provide students with a well-rounded education, it is also crucial to strike a balance between technology and conventional teaching techniques.

## 7. Conclusions and Future Works

The education sector also started to realize the value of technology and how essential it is to the edtech industry. The IoT, or the Internet of Things, is a digital technology that is quickly developing in the education sector. Furthermore, it is successfully improving the current educational system. To maximize technology, the education sector is changing IoT devices and related services. This system helps to make education more dynamic, inclusive, and participatory. Furthermore, it enables interactive learning, guarantees the security of the educational facilities, increases productivity, provides real-time learning opportunities, enables close observation, etc. Additionally, providing your students with IoT devices as teaching aids makes for a truly engaging learning environment. E-learning, which has grown by 900% since 2000 [62], is the sector of the education business with the fastest growth. By 2025, the number of devices using IoT technology is anticipated to reach 75 billion, according to an eLearning Industry Report. This will increase students' interest in taking e-learning courses, because they may profit from these tools by having real-time access to videos and tutorials, for example [63]. The Internet of Things (IoT) is a term that might mean different things to different people, but there is no denying that it incorporates a lot of hardware and data. The International Data Corporation (IDC) forecasts that by 2025, there will be 41.6 billion IoT devices operating worldwide, capable of producing 79.4 zettabytes (ZB) of data [64]. In total, 96% of teachers believe that incorporating emerging technology into the classroom will improve student engagement and learning, based on a current Smoothwall study. A total of 56% of respondents agree that employing technology in the classroom greatly raises student involvement overall [65].

The results of the study show that IoT systems and devices improve access to education. They make teaching and learning easier and democratize education. Students have a tremendously engaging learning experience when IoT devices are used as educational tools. Educational institutions would need to purposefully develop governance structures and strategic plans for integrating the use of this developing technology into institutional life to fully realize the potential of emerging technologies for enhancing teaching and learning processes. Teachers can give grades to students via Telegram and Google Docs while saving time by utilizing Kahoot's online IoT workflows [14]. Real-time measurements of the students' skin resistance, heart rates, and brain waves are carried out using a variety of Internet of Things sensors, including cameras, microphones, and wearable devices [40]. The university intelligence education platform is capable of delivering online instruction, tracking attendance, and querying test results [42]. The Internet of Things Virtual e-learning System (IoT-Ve-LS) paradigm was put up by researchers as a new Smart e-learning Tool (SeT) to continue developing material based on the demands of e-learners [52]. During this COVID-19 epidemic, a blended learning paradigm with an IoT basis would be the best solution for all stakeholders in the education industry [46]. Students gain experience with the learning kit through project-based learning. The "brain", "muscle", and "cloud" components of the kit encompass every aspect of the IoT and blockchain technology [45]. The Internet of Things (IoT) has significant potential to change the face of education. Sig-

nificant accomplishments and ground-breaking research have pushed the limits of what is conceivable in the area of educational technology over time. These developments have improved learning for both students and teachers by making it more individualized, interactive, and data-driven. The IoT has made it possible to create personalized learning experiences, where instructional materials and methods are adjusted to meet the needs of the learner, improving engagement and comprehension. The IoT-enabled smart classrooms with improved temperature, lighting, and air quality make learning more enjoyable and conducive. Now that students can receive immediate feedback on their performance, they may learn from their mistakes and advance more quickly. As a result of the insights into student performance provided by IoT-generated data, educators are now better equipped to make data-driven decisions that are advantageous to both specific students and educational institutions as a whole. The safety of faculty, staff, and visitors has been increased thanks to IoT solutions, which have also improved campus security, emergency response, and transportation safety. Students with impairments now have easier access to education because of the IoT, which has also allowed room for a more welcoming classroom atmosphere.

IoT analytics, wearable technology, online learning and virtual reality, teacher training, and smart campus design are just a few of the areas that have been the subject of pioneering research in IoT-based teaching and learning. Exciting opportunities exist for IoT-based teaching and learning in the future. Personalized learning will be improved by the IoT integration of AI and machine learning, making it possible for more intelligent and adaptive educational systems. Edge computing implementation will lower latency and enhance the real-time capabilities of educational applications in IoT devices. The authentication of educational accomplishments and the issuance of credentials could be made safe, transparent, and impervious to fraud using blockchain technology. Future IoT-based systems should work to further close the digital gap and guarantee that all students have fair access to technological advancements in the classroom. The IoT can promote the idea of lifelong learning by letting people continue to learn new things throughout their lifetimes. The IoT can support international collaboration in education by bringing together students and educators from different cultures. To ensure the ethical and equitable use of these technologies, more study and discussion on the ethical and societal implications of the IoT in education are essential. The IoT has already had a big impact on teaching and learning, and future developments in this area promise education that is more individualized, available, and efficient. The future of IoT-based education will be heavily influenced by pioneering research and ongoing innovation. The study highlights how important it is to address the challenges standing in the way of a smooth Internet of Things (IoT) integration into teaching and learning. The findings demonstrate that integrating the IoT is crucial for achieving teaching and learning goals in the twenty-first century. The study's findings show that employing the IoT in education can help to modernize outmoded materials and techniques utilized in traditional teaching and learning procedures, which frequently elevate the instructor to the role of an authority figure or the only knowledge source. For educators and students to fully benefit from the potential benefits of IoT technology, all parties involved in the education sector must move fast to solve the highlighted obstacles that limit the proper integration of IoT systems and devices into teaching and learning. To estimate future skill gaps and shortages, more research is necessary to discover the talents that are now needed.

## 8. Limitations of IoT-Based Teaching and Learning

The use of technology in teaching and learning has numerous advantages, but certain drawbacks must be considered. Digitalization entails granting unrestricted access to a variety of resources and sources of information, such as websites, social networks, or chats. As a result, they divert attention from the topic. Students who use technology excessively or inappropriately risk developing a compulsive relationship with it, which can make it difficult for them to manage their consumption and hurt their academic, social, and familial lives as well as their health. The extensive use of digitalization in academic

settings may render practices like writing, public speaking, and reasoning useless. Since the younger generations' social abilities are built on the digital world, direct interpersonal communication may be impacted. On the Internet, a lot of the information is inaccurate or lacking. Given that half of students in the ESO educational stage cannot identify fake news, this fact directly affects students' media literacy. Lack of awareness of the risks associated with cybercrime can unintentionally disclose students' data, particularly if they are minors, by exchanging images with random people, for instance. The learning process becomes increasingly asynchronous as new technologies are used, and the amount of face-to-face interaction between students and professors declines. As a result, by limiting social interaction, isolation may develop and cause problems for students' personal growth. Finally, bullying is a difficult topic to address and one of the biggest concerns. Lack of physical touch can result in a loss of assertiveness and overuse of Internet resources, which can result in instances of cyberbullying.

**Author Contributions:** Conceptualization, I.A.G. and M.A.; methodology, I.A.G. and M.A.; software, M.A.; validation, I.A.G. and M.A.; formal analysis, M.A.; investigation, I.A.G. and M.A.; resources, I.A.G. and M.A.; data curation, I.A.G. and M.A.; writing—original draft preparation, M.A.; writing—review and editing, I.A.G. and M.A.; visualization, I.A.G.; supervision, I.A.G. and M.A. All authors have read and agreed to the published version of the manuscript.

**Funding:** The authors extend their appreciation to the Deputyship for Research and Innovation, Ministry of Education in Saudi Arabia for funding this research work through the project number ISP23-75.

**Institutional Review Board Statement:** Not applicable.

**Informed Consent Statement:** Not applicable.

**Data Availability Statement:** The data are available from the corresponding author on reasonable request.

**Conflicts of Interest:** The authors declare no conflict of interest.

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
