# Peer review of "Internet of Things (IoT)-Based Teaching and Learning: Modern Trends and Open Challenges"

_sustainability, doi:10.3390/su152115656_

Round 1

Reviewer 1 Report

Comments and Suggestions for Authors

Dear Authors,

please refer to the attached document. Thanks!

Comments on the Quality of English Language

The manuscript needs to be improved in terms of English language

Author Response

- Although in general, the aim of the paper is clear, the abstract should be elegant and well-written highlighting points as the novelty respect the state of art. It is opinion of this reviewer that the abstract requires a restructuring to have a better fluency. It should be better organized. This will assist the readers quickly to understand the pros and cons of this model.

The abstract has been revised, lines 8 to 25.

- The keywords are appropriate. However, this is only a reviewer's opinion, it is better to use keywords that better highlight the focus of the paper. Authors have the possibility to insert a maximum of five keywords and no eight keywords.

The keywords have been revised, line 26.

- with the aim of improving the architecture of the paper, would be more appropriate to insert at the end of the “introduction section” the expression: “The architecture of the paper is organized as follows:…………. and to indicate section 1……..section 2…….. and so on”.

This section has been added at the end of the introduction as per reviewer advice, lines 163 to 169.

- The authors need to mention and highlight what the current paper adds to the discussion, and in what way it is different from other reviews in this field. In other words, highlight the novelty.

This comment has been incorporated in the revised version, lines 549 to 582.

- Concepts are accessible for the non-specialist in this field. However, this reviewer does not understand if this manuscript is a mini-review seeing that this paper seems to summarize the most salient concepts related to a topic while reporting the most relevant and current findings or a review which should include figures, diagrams, and tables, explanatory and consistent with the text itself. Specify it in the text.

This comment has been incorporated throughout the revised version.

- In your interest, in the case of review manuscript please, you should put figures, tables, diagrams in each section of the review. They are need in order to make the review more interesting for the reader.

Figures have been added as per reviewer advice, figures 1, 2, 3, 4, 5, and 6.

- The potential impact of this work should be better presented in the manuscript.

The potential impact of this research work has been added, from lines 158 to 163.

- The number of recent articles is very limited in case the article read is a review. Some following references are good for your paper if you wish to consider them.

The recent articles and suggested references have been added in the revised version, lines 444 to 472. Reference 3 and 5.

- Please, highlight the strength, weaknesses, opportunities and threats were derived from yours study.

Table 2 summarizes the SWOT analysis of the study.

- The manuscript briefly mentions the use of advanced devices, sensors, and software in Teaching and Learning systems. It would be helpful to elaborate on how these components contribute to improvement the student performance.

This comment has been incorporated in the revised version, lines 587 to 610.

- It would be beneficial to provide a more in-depth analysis of the specific challenges and potential solutions associated with IoT for Teaching and Learning systems.

A more in-depth analysis of the challenges and potential solutions has been incorporated in the revised version, lines 517 to 545.

-It is essential to discuss the limitations and potential drawbacks based on the Internet of Things.

The limitations of IoT-based teaching and learning have been added in the revised version, lines 699 to 719.

- The conclusion section should include also the main achievements and pioneering research as well as mention future directions more tangibly.

The conclusion has been revised as per reviewer advice, lines 650 to 686.

- The manuscript needs to be improved in terms of English language.

Professional proofreading has been done before submitting the revised version.

Reviewer 2 Report

Comments and Suggestions for Authors

The introduction, study methods, and conclusion sections of this paper are nicely written and include scholarly discussion. Additionally, the authors go into very little detail on current IOT-based teaching and learning approaches. Some recent publications are included in this area as my advice to authors.

Author Response

- The introduction, study methods, and conclusion sections of this paper are nicely written and include scholarly discussion. Additionally, the authors go into very little detail on current IOT-based teaching and learning approaches. Some recent publications are included in this area as my advice to authors.

The recent articles have been added in the revised version, lines 444 to 472.

Reviewer 3 Report

Comments and Suggestions for Authors

The abstract references the use of bibliometric analysis with VOSviewer to examine research articles but does not provide any specific findings from this analysis. Consider incorporating some high-level findings or trends to showcase the relevance and significance of the research.

While the manuscript discusses the benefits of IoT in education, it doesn't appear to address potential drawbacks or criticisms. Including a balanced perspective by mentioning potential challenges or limitations of IoT integration in education would enhance the credibility of the article.

While the introduction refers to previous studies and research findings, it lacks specific citations to support these references. Incorporate proper citations to academic sources to strengthen the credibility of your statements and provide readers with opportunities to explore the referenced research.

Some sentences are complex and convoluted, making it challenging for readers to grasp the intended meaning

Author Response

- The abstract references the use of bibliometric analysis with VOSviewer to examine research articles but does not provide any specific findings from this analysis. Consider incorporating some high-level findings or trends to showcase the relevance and significance of the research.

This comment has been incorporated in the revised version, lines 270 to 276.

- While the manuscript discusses the benefits of IoT in education, it doesn't appear to address potential drawbacks or criticisms. Including a balanced perspective by mentioning potential challenges or limitations of IoT integration in education would enhance the credibility of the article.

The more in-depth analysis of the challenges and potential solutions has been incorporated in the revised version, lines 517 to 545.

The limitations of IoT-based teaching and learning have been added in the revised version, lines 699 to 719.

- While the introduction refers to previous studies and research findings, it lacks specific citations to support these references. Incorporate proper citations to academic sources to strengthen the credibility of your statements and provide readers with opportunities to explore the referenced research.

All references are properly cited as per reviewer comments.

- Some sentences are complex and convoluted, making it challenging for readers to grasp the intended meaning.

Professional proofreading has been done before submitting the revised version.

Round 2

Reviewer 1 Report

Comments and Suggestions for Authors

Dear Authors,

please refer to the attached document. Thanks!

Reviewer 3 Report

Comments and Suggestions for Authors

All comments are well addressed and so accepted as it is.

Comments on the Quality of English Language

Acceptable